

# Validation study on the assay method for anti-factor IIa potency of enoxaparin sodium

Xiaorong Yang, Hanyan Zou, Yixue Dong, Bing Liu, Ying Wang and Mengying Wang

Chongqing Institute for Food and Drug Control, Chongqing, China

## ABSTRACT

Enoxaparin sodium is a low molecular mass heparin essential for effective anticoagulation therapy. However, significant variations in testing methods across different manufacturers have led to poor reproducibility of results, increasing the risks associated with drug quality evaluation by manufacturers and regulatory oversight. This study integrates the strengths of various testing methods to establish a reproducible assay that has been thoroughly validated. The validation results demonstrate that the method exhibits excellent specificity, linearity, robustness, precision, and accuracy, with recovery rates ranging from 98.0% to 102.0%. The new method demonstrated high consistency and reproducibility, with an RSD value of less than 2.0%, and showed the potential to replace the European Pharmacopoeia method by reducing reagent usage, experimental costs, and equipment requirements. The reliable results of this method facilitate its adoption across different laboratories, enhance the quality control of enoxaparin sodium, and provide a reference for new manufacturers and drug regulatory authorities, thereby ensuring medication safety.

## INTRODUCTION

Enoxaparin sodium is the sodium salt that consists of a complex mixture of oligosaccharides with low molecular mass (*Council of Europe, 2014*), representing a type of low molecular mass heparin sodium (*Council of Europe, 2021*) with a mass-average relative molecular mass ranging between 3,800 and 5,000, characterized by a value of about 4,500. The United States Pharmacopoeia, the Chinese Pharmacopoeia, and the European Pharmacopoeia (EP) similarly focus on the activity of anti-factor Xa and anti-factor IIa. The ratio of anti-factor Xa to anti-factor IIa activity is used as a molecular weight evaluation parameter for heparin-based biologic drugs. According to the requirements for product quality review in the Good Manufacturing Practice (GMP) guidelines for medical products, manufacturers must conduct in-batch stability studies and inter-batch consistency assessments for key product parameters. These evaluations assess the stability and reliability of drug quality during production. The anti-factor IIa activity of enoxaparin sodium, primarily achieved by inhibiting thrombin (factor IIa) (*Vølund, 1978*), is one of

Corresponding author
Xiaorong Yang,
yangXiaorong@cqifdc.org.cn

the key parameters for the quality assessment. Throughout the drug's lifecycle, this parameter must be evaluated for each batch, with multiple tests conducted across various laboratories to ensure its consistency and reliability. The robust, reproducible, and reliable testing method is essential to facilitate laboratory testing by various manufacturing companies and regulatory laboratories, ensuring consistent evaluation and oversight.

However, existing testing methods have been developed by various enoxaparin sodium manufacturers or are based on the potency testing of low molecular mass heparins, as outlined in the EP (No. 07/2021:0828). These methods are influenced by factors such as differing equipment, variations in laboratory personnel, and the sources of reagents, leading to inconsistencies in testing procedures. This variability results in poor reproducibility between laboratories, thereby increasing the risks associated with medication use. This study aims to establish a robust and operable testing method that can be performed by a single operator without the need for specialized instruments, by integrating testing methods provided by multiple companies. This approach seeks to enhance reproducibility across different laboratories. The method involves using internationally recognized standard samples with known potency to calibrate the potency of the analytes in heparin assays. The principle of detection is based on calibrating the potency of the test samples using the values of the standard references, and the method undergoes comprehensive validation according to the Analytical Method Validation Guidelines of the General Principles 9,101 of the Chinese Pharmacopeia, 2020 edition (*Commission, 2020*). The validation results are satisfactory, indicating that this method can be effectively utilized to assess its reliability and feasibility.

# METHODS

## Reagents and materials

The low molecular mass heparin Biological Reference Preparation, batch number 11, manufactured by the European Directorate for the Quality of Medicines & HealthCare (EDQM), possesses an anti-factor IIa activity of 37 IU/mL. It is specifically designed for use in the anti-factor IIa assay as detailed in the Ph. Eur. monograph on heparins, low molecular mass (0828) (*Commission, 2020*; *Pei-hong, Ying-zi & Rong-fu, 2020*).

The Thermo Varioskan Flash, an ultraviolet-visible spectrophotometer, was utilized as a reader for the enzyme-linked immunosorbent assay (ELISA).

The Shimadzu UV-2600, an ultraviolet-visible spectrophotometer, was used to measure the absorbance of the mixed solution using semi-micro cuvettes.

The antithrombin III solution R2 (*Council of Europe, 2010a*), procured from BioWill with batch number 230,823 and containing 10 IU per bottle, was diluted to achieve a concentration of 0.5 IU/mL using a Tris(hydroxymethyl)aminomethane sodium chloride buffer solution pH 7.4 R (pH 7.4) solution.

The human thrombin solution R (*Council of Europe, 2010c*), obtained from HYPHEN BioMed with batch number FB0044 and containing 100 IU per bottle, was diluted to a final concentration of 5 IU/mL using a *pH 7.4* solution.

The chromogenic substrate R2 S-2238 (*Council of Europe, 2010b*), provided by BioWill with batch number 230,317 and weighing 10 mg per bottle, was initially diluted to a

concentration of 3 mM using purified water R. Prior to utilization, it was further diluted to a concentration of 0.5 mM with a pH 8.4 buffer solution.

A 30% of acetic acid solution was prepared by diluting 30 milliliters of glacial acetic acid to a total volume of 100 milliliters in a volumetric flask using purified water R.

The manufacturer of enoxaparin sodium is Yino Pharma Limited, and the three batches of samples are YN001, YN002, and YN003. Based on the preliminary experiment results, the estimated anti-factor IIa potency of the samples is 26 IU/mg, which was used for calculations during solution dilution (Table 1). Among these, YN001 is used for methodological validation. After confirming the method's feasibility, YN001, YN002, and YN003 will be tested.

## Solution preparation

The dilution scheme for the preparation of standard and test solutions is conducted according to Tables 1 and 2. The validation experiments for specificity, linearity, and range use the following stock solutions: pH 7.4 solution, six standard solutions ($SC_{Max}$, $SC_{Min}$, $S_1$, $S_2$, $S_3$, and $S_4$), and test solution $T_4$. A total of nine solutions—pH 7.4 solution, four standard solutions ($S_1$, $S_2$, $S_3$, and $S_4$), and four test solutions ($T_1$, $T_2$, $T_3$, and $T_4$)—are used as test solutions for other validation experiments. Enoxaparin sodium, the solid test sample, has an estimated concentration that varies with each sample weight. The estimated concentration of test sample T is calculated as follows:

$$\text{Estimated activity of test Sample T(IU/mL)} = \frac{\text{sample weight (mg)} \times \text{estimated potency (IU/mg)}}{\text{dilution volume (mL)}}$$

If the measured potency of anti-factor IIa exceeds the estimated value by more than ±10%, the potency is re-estimated, the sample is re-diluted, and retesting is performed.

## Assay method and potency calculation

### Tests to be evaluated

A 96-well plate is used, with the test solutions arranged in a square grid, and 16 µL of each test solution is added. Using a multi-channel pipette, the following reagents are added sequentially at the same speed and frequency, with thorough mixing after each addition. Then, 16 µL of antithrombin III solution is added to each well, followed by incubation at 37 °C for 1 min. A total of 32 µL of human thrombin solution is added, and the plate is incubated again at 37 °C for 1 min. Subsequently, 80 µL of chromogenic substrate solution is added, and the plate is incubated for 4 min at 37 °C. The reaction is stopped by adding 120 µL of 30% acetic acid solution. The absorbance at 405 nm is measured using an ELISA reader.

### European pharmacopoeia potency testing as control tests

A 2.0 mL microcentrifuge tubes with a round bottom are used, and 50 µL of each test solution and 50 µL of antithrombin III solution are added to each tube, followed by incubation at 37 °C for 1 min. Next, 100 µL of human thrombin solution is added, and the

**Table 1 Preparation of series test solutions.**

| ID | Estimated activity (IU/mL) | Be diluted (mL) | | | Diluents (mL) | |
|---|---|---|---|---|---|---|
| Test Sample T | 24.04* | / | | | | |
| $TC_0$ | 3.7 | Test Sample T | 0.179* | + | Water R | 1.0 |
| $TC_1$ | 1.0 | $TC_0$ Solution | 0.2 | + | Water R | 0.54 |
| $TC_2$ | 0.1 | $TC_1$ Solution | 0.2 | + | pH7.4 | 1.8 |
| $T_1$ | 0.065 | $TC_2$ Solution | 0.65 | + | pH7.4 | 0.35 |
| $T_2$ | 0.0455 | $T_1$ Solution | 0.7 | + | pH7.4 | 0.3 |
| $T_3$ | 0.03185 | $T_2$ Solution | 0.7 | + | pH7.4 | 0.3 |
| $T_4$ | 0.022295 | $T_3$ Solution | 0.7 | + | pH7.4 | 0.3 |

Notes:
* The sample weight of the test substance may vary slightly during sample preparation.
Abbreviations: pH 7.4, tris(hydroxymethyl)aminomethane sodium chloride buffer solution pH 7.4 R.

**Table 2 Preparation of series standard solutions.**

| ID | Labeled activity (IU/ml) | Be Diluted (mL) | | | Diluents (mL) | |
|---|---|---|---|---|---|---|
| Standard S | 37.0 | 1.0 ml of water R for reconstitution | | | | |
| $SC_0$ | 3.7 | Standard S | 1.0 | + | Water R | 9.0 |
| $SC_1$ | 1.0 | $SC_0$ Solution | 0.2 | + | Water R | 0.54 |
| $SC_2$ | 0.1 | $SC_1$ Solution | 0.2 | + | pH7.4 | 1.8 |
| $S_1$ | 0.065 | $SC_2$ Solution | 0.65 | + | pH7.4 | 0.35 |
| $S_2$ | 0.0455 | $S_1$ Solution | 0.7 | + | pH7.4 | 0.3 |
| $S_3$ | 0.03185 | $S_2$ Solution | 0.7 | + | pH7.4 | 0.3 |
| $S_4$ | 0.022295 | $S_3$ Solution | 0.7 | + | pH7.4 | 0.3 |
| $SC_{Max}$ | 0.09 | $SC_2$ Solution | 0.09 | + | pH7.4 | 0.01 |
| $SC_{Min}$ | 0.012 | $SC_2$ Solution | 0.012 | + | pH7.4 | 0.088 |

tubes are incubated again at 37 °C for 1 min. Subsequently, 250 μL of chromogenic substrate solution is added, and the tubes are incubated for 4 min at 37 °C. The reaction is stopped by adding 375 μL of 30% acetic acid solution. The mixtures are then transferred to semi-cuvettes, and absorbance is measured using a Shimadzu UV-2600 spectrophotometer at 405 nm.

## Potency calculation

The assay used in this experiment is based on a chromogenic substrate method. In this approach, the analyte reacts sequentially with purified coagulation factors, antithrombin III and human thrombin, to form a complex. Following the reaction, excess human thrombin quantitatively hydrolyzes the specific chromogenic substrate R2 S-2238, resulting in color development (*Al-Sallami & Medlicott, 2015*). The reaction is terminated using an acetic acid solution, and the absorbance of the endpoint solution is measured. The absorbance is inversely proportional to the concentration of the analyte. A linear regression is performed with absorbance as the dependent variable and the logarithm of the concentration of the standard or test solutions as the independent variable. The

potency is calculated using the principle of the 4 × 4 parallel line model (*Vølund, 1978*). The potency of the test substance per 1 mg of anti-factor IIa, considering the moisture loss of 3.9%, is calculated.

## RESULTS

### Specificity, linearity, and range

Using the assay method described above, a pH 7.4 buffer solution was used as a blank control (B). The absorbance of the solutions listed in Tables 1 and 2, including $SC_{Max}$, S1, S2, S3, S4, $SC_{Min}$, and T4, was measured. The results are presented in Table 3 and include the following calculations:

The absorbance of the blank control was compared to that of the other solutions. The blank control had higher readings than the other solutions, indicating good specificity.

A linear equation was fitted for absorbance against the logarithm of concentration, and the correlation coefficient $R$ was calculated. A correlation coefficient of $R = 0.9996$ indicates that the absorbance is linearly related to the solution concentration within the range of 0.012 to 0.09 IU/mL.

### Robustness

Robustness was assessed to determine the degree to which the assay results are unaffected by small variations in the testing conditions. This experiment primarily investigated the effects of variations in incubation temperature, incubation time, and the duration for which the test sample solutions are left standing. Variations in incubation temperature involved measuring the results at 39 °C (P2) and 35 °C (P3) compared to 37 °C (P0). Reaction time variations involved extending or reducing each reaction step's incubation time at 37 °C by 15 s (P1 and P4, respectively). The effect of standing time for the sample solution was assessed by repeating dilution and testing after the test sample solution had been left at room temperature for 24 h.

The potency of the test samples was determined by varying the incubation temperatures and times. Results showed that changes in incubation temperature to 39 °C or 35 °C, or changes in incubation time by an increase or decrease of 15 s, yielded potencies with RSD values ranging from 1.4% to 4.4% (see Table 4). Potency measured after the test sample solution stood for 24 h (P24hr) compared to the potency of the freshly prepared solution measured on the same day (P0hr) showed an RSD value of less than 2.0% (see Table 4).

### Repeatability and intermediate precision

Repeatability: The potency determination of six test sample solutions shows an RSD value of less than 2.0%, as seen in Table 5.

Intermediate precision: This involves another operator measuring the potency of six test sample solutions. The results showed an RSD value of less than 2.0%, and compared with the repeatability of the six test sample potencies, the RSD is less than 2.0%, as seen in Table 5.

Reproducibility: Two independent laboratories tested the anti-factor IIa factor potency in three batches of enoxaparin sodium, yielding consistent results, as shown in Table 6.
**Table 3 Results of specificity, linearity, and range.**

| Solution | Labeled activity (IU/mL) | Absorbance | Comparison | Correlation coefficient R |
|---|---|---|---|---|
| B | / | 1.0027 | / | / |
| $SC_{Max}$ | 0.09 | 0.3292 | / | $y = -0.59609 \times -0.29775$ |
| $S_1$ | 0.065 | 0.4058 | / | R = 0.9996 |
| $S_2$ | 0.0455 | 0.4974 | / | |
| $S_3$ | 0.03185 | 0.5958 | / | |
| $S_4$ | 0.022295 | 0.6946 | $B > S_4$ | |
| $SC_{Min}$ | 0.012 | 0.8435 | $B > SC_{Min}$ | |
| $T_4$ | / | 0.7059 | $B > T_4$ | / |

**Table 4 Robustness results.**

| ID | Measured activity (IU/mg) | Dry matter potency (IU/mg) | RSD compared to P0 |
|---|---|---|---|
| $P_0$ (Initial) | 26.691 | 27.77 | / |
| $P_1$ (+15 s) | 27.368 | 28.48 | 1.8% |
| $P_4$ (−15 s) | 25.081 | 26.941 | 4.4% |
| $P_2$ (+2 °C) | 27.378 | 28.49 | 1.8% |
| $P_3$ (−2 °C) | 27.213 | 28.32 | 1.4% |
| P0hr | 26.037 | 27.01 | RSD compared to P0hr |
| P24hr | 25.948 | 27.00 | 0.2% |

**Table 5 Results and analysis of repeatability and intermediate precision (activity units: IU/mg).**

| ID | | Weighing value (mg) | Estimated activity | Measured activity | Dry substance activity | RSD repeatability | RSD intermediate precision |
|---|---|---|---|---|---|---|---|
| Repeatability | Re1 | 50.02 | 25.98 | 25.859 | 26.91 | 1.8% | 1.7% |
| | Re2 | 50.43 | 25.77 | 24.948 | 25.96 | | |
| | Re3 | 49.64 | 26.18 | 25.517 | 26.55 | | |
| | Re4 | 51.10 | 25.43 | 25.164 | 26.19 | | |
| | Re5 | 50.39 | 25.79 | 25.419 | 26.45 | | |
| | Re6 | 49.87 | 26.06 | 26.22 | 27.28 | | |
| Intermediate Precision | Re1 | 51.57 | 26.01 | 24.893 | 25.90 | 1.1% | |
| | Re2 | 50.03 | 25.97 | 25.003 | 26.02 | | |
| | Re3 | 50.65 | 25.66 | 25.569 | 26.61 | | |
| | Re4 | 50.77 | 25.59 | 24.821 | 25.83 | | |
| | Re5 | 50.17 | 25.90 | 25.098 | 26.12 | | |
| | Re6 | 50.40 | 25.78 | 24.919 | 25.93 | | |

**Table 6  Detection results of two independent laboratories on three batches of test samples.**

| Anti-factor IIa activity (IU/mg) | YN001 | YN002 | YN003 |
|---|---|---|---|
| Laboratory 1 | 25.5 | 25.2 | 25.7 |
| Laboratory 2 | 27.1 | 27.2 | 25.8 |

**Table 7  Accuracy of the prepared test samples and results (activity Units: IU/ml).**

| ID | Measured activity | Standard added value | Measured activity After addition | Recovery rate | RSD |
|---|---|---|---|---|---|
| SP1 (Initial Sample 1) | 25.948 IU/mg | / | 26.69 IU/ml | / | 1.4% |
| PB1 (Spiking 1) | / | 25.99 IU/ml | 26.122 IU/ml | 98.3% | |
| PB2 (Spiking 2) | / | 25.99 IU/ml | 26.21 IU/ml | 99.0% | |
| PB3 (Spiking 3) | / | 25.99 IU/ml | 26.119 IU/ml | 98.3% | |
| PB4 (Spiking 4) | / | 25.99 IU/ml | 26.351 IU/ml | 100.1% | |
| PB5 (Spiking 5) | / | 25.99 IU/ml | 26.562 IU/ml | 101.7% | |
| SP2 (Initial Sample 2) | 25.446 IU/mg | / | 26.34 IU/ml | / | |
| PB6 (Spiking 6) | / | 25.99 IU/ml | 26.307 IU/ml | 101.1% | |

## Accuracy

Accuracy was assessed by adding a known concentration of the standard to the test sample solutions and calculating the recovery rate. The results showed that the recovery rates for the test sample solutions ranged between 98.0% and 102.0%, indicating good accuracy. The experiment was repeated six times, with the calculated recovery rate's RSD being less than 2.0%, as shown in Table 7.

## Comparison with control test results

This involves two operators working in close coordination to measure the potency of six test sample solutions. The results showed an RSD value of less than 2.0%. When compared to the repeatability of the six test sample potencies, the RSD was also below 2.0%, as shown in Table 8.

## DISCUSSION

This experiment validated the accuracy and reliability of the method for determining the anti-factor IIa potency in enoxaparin sodium. The results demonstrated that the method meets the guidelines of the Chinese Pharmacopoeia in terms of specificity, linearity and range, robustness, precision, and accuracy. Notably, regarding robustness, even with minor changes in experimental conditions (such as temperature adjustments or variations in reaction times), the method showed a low relative standard deviation (RSD), indicating good robustness. Consistency was high among different operators, and the results were consistent between two independent laboratories, suggesting high method stability, facilitating method comparison and transfer across different laboratories. Additionally, the validation of precision and accuracy confirmed the method's reliability. These results are

**Table 8 Results and analysis of repeatability and control tests from the european pharmacopoeia (activity units: IU/mg).**

| ID | | Weighing value (mg) | Estimated activity | Measured activity | Dry substance activity | RSD repeatability | RSD intermediate precision |
|---|---|---|---|---|---|---|---|
| Repeatability | Re1 | 50.02 | 25.98 | 25.859 | 26.91 | 1.8% | 1.9% |
| | Re2 | 50.43 | 25.77 | 24.948 | 25.96 | | |
| | Re3 | 49.64 | 26.18 | 25.517 | 26.55 | | |
| | Re4 | 51.10 | 25.43 | 25.164 | 26.19 | | |
| | Re5 | 50.39 | 25.79 | 25.419 | 26.45 | | |
| | Re6 | 49.87 | 26.06 | 26.22 | 27.28 | | |
| Control tests | Re1 | 50.38 | 25.79 | 25.593 | 26.63 | 1.4% | |
| | Re2 | 50.49 | 25.74 | 26.159 | 27.22 | | |
| | Re3 | 49.74 | 26.13 | 26.441 | 27.51 | | |
| | Re4 | 51.71 | 25.13 | 25.66 | 26.70 | | |
| | Re5 | 50.32 | 25.82 | 26.088 | 27.15 | | |
| | Re6 | 50.35 | 25.81 | 26.392 | 27.46 | | |

significant for ensuring the quality control of enoxaparin sodium. To further ensure the reliability of the results in this study, we conducted a comparative analysis with the method outlined in the EP heparins, low-molecular-mass (No. 07/2021:0828). The results showed good consistency, indicating that the new method can replace the EP method. Compared to the EP method, the method described in this study requires only one-third of the test sample and reagent volumes, thus reducing experimental costs while decreasing the overall volume of the reaction system. This allows the experiment to be efficiently conducted in a 96-well plate using a multi-channel pipette, rather than performing the experiment in microcentrifuge tubes and transferring the reaction solutions individually to semi-cuvettes for absorbance measurement. The method reported in this study can be performed by a single operator using a multi-channel pipette, eliminating the need for collaboration between two operators and costly equipment. Because this method is highly operable, a coagulation analyzer is not required for the experiment.

The findings of this study are consistent with existing literature on the potency testing of low molecular mass heparins. For example, other studies have also found that the potency of anti-factor IIa can be accurately determined through the linear relationship between absorbance and concentration (*Zhuo-wei et al., 2022*). Moreover, the robustness and precision results from this study align with those of similar studies, further confirming the stability of this testing method.

The standard used in this experiment, heparin low molecular mass for biological reference preparation, is an accredited standard by the European Pharmacopoeia (Ph. Eur.). The key reagents, antithrombin III solution and chromogenic substrate, were sourced from BioWill, while the human thrombin solution was obtained from HYPHEN BioMed. All reagents are lyophilized and each kit contains 10–12 vials, demonstrating good stability when stored in a 2–8 °C environment, with a shelf life of up to 24 months. Additionally, the national standard of blood coagulation factor VIII concentrate, factor IIa (thrombin), and chromogenic

substrate S-2238 from the National Drug Reference Standards, which are sold as single vials, allowing flexible purchasing based on actual requirements and minimizing unnecessary reagent waste.

Despite these supportive findings, there are some limitations to the study. Firstly, the sample size for inter-laboratory comparison was small (only three data points per lab), which may limit the statistical power of the results. Secondly, the strict control of experimental conditions might be challenging to replicate entirely in practical applications, which could affect the consistency of the test results.

## CONCLUSION

The validation of the method for determining anti-factor IIa potency in enoxaparin sodium was successful, providing an efficient, low-cost, and highly operable testing method for enoxaparin sodium manufacturers and regulatory authorities. This method serves as an objective and accurate evaluation tool for the quality control of enoxaparin sodium, ensuring medication safety. Furthermore, the methodology and results of this study are of referential value for the establishment and validation of quality standards for other similar drugs.

### Funding

This work was supported by the project of Chongqing Science and Technology Commission (No. 2018004). The funders had no role in study design, data collection and analysis, decision to publish, or preparation of the manuscript.

### Grant Disclosures

The following grant information was disclosed by the authors:
Chongqing Science and Technology Commission: 2018004.

### Competing Interests

Yang Xiaorong, Zou Hanyan, Dong Yixue, Liu Bing, Wang Ying and Wang Mengying are employed by Chongqing Institute for Food and Drug Control.

### Author Contributions

- Xiaorong Yang conceived and designed the experiments, performed the experiments, prepared figures and/or tables, authored or reviewed drafts of the article, and approved the final draft.
- Hanyan Zou analyzed the data, authored or reviewed drafts of the article, and approved the final draft.
- Yixue Dong performed the experiments, prepared figures and/or tables, and approved the final draft.
- Bing Liu analyzed the data, prepared figures and/or tables, and approved the final draft.
- Ying Wang analyzed the data, authored or reviewed drafts of the article, and approved the final draft.

- Mengying Wang performed the experiments, analyzed the data, prepared figures and/or tables, and approved the final draft.

## Data Availability

The raw measurements are available in the Supplemental File.

## Supplemental Information

Supplemental information for this article can be found online at http://dx.doi.org/10.7717/peerj.18732#supplemental-information.

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
