# Peer review of "Validation study on the assay method for anti-factor IIa potency of enoxaparin sodium"

_PeerJ, doi:10.7717/peerj.18732_

## Round 0.1 · original submission · Major Revisions

Please address issues pointed by the reviewers and amend manuscript accordingly.

Reviewer 1 ·

Basic reporting

The paper reports the validation study on Anti-IIa potency assay of enoxaparin sodium. Anti-IIa potency assay of enoxaparin sodium is well an established standard assay. The paper seems a technical report rather than a research paper.

Experimental design

The experimental design seems just followed the Anti-IIa potency assay protocol.

Validity of the findings

No novelty with less impact.

Reviewer 2 ·

Basic reporting

The language and English is good and clear.
However the aim for this antiIIa assay as compared to the ones that are already commercially available is not clear.
The Table Legends should explain the abbreviations used

Experimental design

The experimental design follows the standard laboratory standards and hence it is good. Again the research question is unclear to me. Are the authors validating their own method with their own reagents? This is what it looks like.
The paper is lacking of detail of manufacturers of the different reagents used and hence it is not easily replicated by other labs

Validity of the findings

The validity fails in many ways. The most important are:
1. There is no comparison with currently available AntiIIa assays. These are commercially available and hence there should not have been issue procuring them
2. There are no ex vivo experiments. It seems that the authors performed spiking experiments but they did not extend it to samples from patients since these might sometimes behave differently
3. To me it seems that only the Standard LMWH solution was used and no detailed validation using different enoxaparins apart from the small interlaboratory exercise. Also the type of enoxaparin sodium ie manufacturer etc is lacking.

Additional comments

More work needs doing to confirm validity of this assay especially comparing it to currently available antiIIa assays

Reviewer 3 ·

Basic reporting

no comment

Experimental design

no comment

Validity of the findings

The authors of the manuscript titled “Validation study on the assay method for Anti-IIa potency of enoxaparin sodium” present a validation study of an anti-IIa assay for enoxaparin.
This is an interesting study but the rationale and implications are not clear to me. I have a few questions and for the authors to consider.

Main comments:
1. Why do we need a potency assay for enoxaparin? Is this in relation to assessing enoxaparin bio-similars? Please discuss the literature gap and rationale for this work.
2. Why do you need to assay anti-IIa with enoxaparin? Anti-Xa is an established assay for enoxaparin potency and also clinical use.
Please note this published work (https://doi.org/10.1111/jphp.12333) where an enoxaparin anti-Xa assay was validated in terms of robustness, extended expiry, sample haemolysis, etc. Can your work provide provide external validity for anti-IIa assay?
3. Why was the anti-IIa range chosen? Typical anti-Xa range is 0.1-1.1 IU/L?
4. The manuscript has a few grammatical and typographical errors that affect the clarity and flow of the work. Please enlist the help of a professional proofreading service.

Regards and best wishes.

---

## Round 0.2 · Major Revisions

As you can see, one of the original reviewers (R2) is dissatisfied by your responses and revision and once again recommend rejection. I am giving you another chance to adequately address their concerns and amend manuscript accordingly.

Reviewer 2 ·

Basic reporting

See comments in 4 below

Experimental design

See comments in 4 below

Validity of the findings

See comments in 4 below

Additional comments

Unfortunately the authors fail to provide a comparative analysis against existing test kits and hence validity of this new assay cannot be assessed
The authors also seem to infer that the Anti IIa activity of enoxaparin is the most important in a similar drug which is incorrect. It is its antiXa activity which is the most important and used for dosing

Reviewer 3 ·

Basic reporting

pls see below

Experimental design

pls see below

Validity of the findings

pls see below

Additional comments

The authors have responded to my review adequately and have made tracked changes accordingly. However, some statements are either inaccurate or not relevant. Please consider rephrasing:
- Re anti-Xa assay: the purpose of this study is to validate an anti-IIa assay in order to standardise the quality assay used by manufacturers of enoxaparin. Clinically, patient response to enoxaparin is assessed using anti-Xa assay (not anti-IIa). This is because enoxaparin targets factor Xa much more potently than IIa. This is not a problem for this study, and anti-IIa and anti-Xa are highly correlated, you just need to make it clear that this study is about assessing enoxaparin manufacturing quality rather than clinical use.
- Re enoxaparin administration: the drug is given subcutaneously and IV use is unnecessary and discouraged. This is, again, not a problem for this study but the link between subcut use and quality makes little sense to me. Also, the link to thrombocytopenia is not relevant, unless you are alluding to a link between thrombocytopenia and the manufacturing quality of enoxaparin?
I think you need to steer away from the clinical use of enoxaparin and focus on manufacturing quality, drug registration, and assay validation.

Regards and best wishes.

---

## Round 0.3 · Minor Revisions

Please address remaining issues pointed by the reviewer and amend manuscript accordingly.

Reviewer 3 ·

Basic reporting

Pls see under Comments

Experimental design

Pls see under Comments

Validity of the findings

Pls see under Comments

Additional comments

• The authors have now revised this manuscript twice. They have addressed most of the issues raised by me and the other reviewers. However, the authors have written new text that is still wrong or misleading. Please steer away from enoxaparin clinical use as it's clearly outside of your expertise. Two clear examples:
o “Enoxaparin Sodium has a wide range of approved indications, multiple routes of administration…”
This is incorrect. Although intravenous and subcutaneous routes were approved initially, only the subcutaneous route is now indicated in standard datasheets and clinical guidelines.
o “The Anti-factor IIa activity of enoxaparin sodium is one of a critical parameter for dosing.”
This is incorrect. Only anti-Xa activity is assessed in dosing in clinical practice.

• A minor point, please don’t capitalise generic drug names. “Enoxaparin Sodium” should be “enoxaparin sodium” or “Enoxaparin sodium” if appears at the beginning of a new sentence.

Regards and best wishes.

---

## Round 0.4 · accepted · Accept

All issues pointed by the reviewers were adequately addressed and the revised manuscript is acceptable now